# Medication error reporting in Ghana: A multicenter assessment of healthcare professionals' knowledge, attitudes and practices

Stephen Mensah Arhin[1]*, Isaac Tabiri Henneh[2], Kwasi Sobre Nkrumah[3], Ewura Seidu Yahaya[1], Phyllis Elsie Owusu Agyei[1], Meshack Antwi-Adjei[1], Martins Ekor[1]

1 Department of Pharmacology, School of Medical Sciences, University of Cape Coast, Cape Coast, Ghana, 2 Department of Pharmacotherapeutics and Pharmacy Practice, School of Pharmacy and Pharmaceutical Sciences, University of Cape Coast, Cape Coast, Ghana, 3 Department of Internal Medicine and Therapeutics, School of Medical Sciences, University of Cape Coast, Cape Coast, Ghana

* stephen.arhin@ucc.edu.gh

## Abstract

### Background

Medication errors (MEs) remain a leading cause of preventable patient harm globally, with significant implications in low- and middle-income countries (LMICs) where reporting systems are not standardized. Understanding healthcare professionals' knowledge, attitudes and practices is critical to designing interventions that strengthen patient safety.

### Methods

A cross-sectional, multicenter study was conducted among healthcare professionals, including nurses, doctors, and pharmacists. A structured questionnaire assessed sociodemographic characteristics, knowledge of categories and sources of MEs, medication administration practices, and awareness and use of error reporting systems. Data were analyzed using descriptive statistics and Poisson regression with robust standard errors to identify factors associated with reporting system awareness.

### Results

A total of 2078 healthcare professionals including nurses, doctors, and pharmacists were surveyed. Knowledge levels varied across different medication error categories, with highest recognition for wrong dose (85.5%) and wrong patient errors (81.5%), while shortage of drugs showed lower recognition (44.0%). While adherence to some safety practices was strong (86.6% always checked patient identity), unsafe behaviors such as preparing medications for more than one patient at a time (37.4%)

**Data availability statement:** Data access is restricted by the Institutional Review Board of the University of Cape Coast in accordance with participant confidentiality protections and ethical approval requirements. Qualified researchers may request access to the minimal anonymized dataset by contacting UCCIRB at irb@ucc.edu.gh. Data will be made available on reasonable requests and with approval from the relevant institutional ethics body.

**Funding:** The study was funded by the Directorate of Research, Innovation and Consultancy, University of Cape Coast (DRIC-UCC), with grant number DRIC-RSG-PPO-24-002. The funders had no role in study design, data collection and analysis, decision to publish, or preparation of the manuscript.

**Competing interests:** The authors declare that no competing interests exist.

and failure to double-check insulin doses (only 33.9% compliance) were reported. Although 62.6% were aware of institutional error reporting systems, only 23.6% consistently reported errors, and over 53% were uncertain about the process of submitting incident forms.

## Conclusion

Healthcare professionals in Ghana demonstrate strong knowledge of medication error categories and sources but notable gaps persist in safe medication practices and reporting behaviors. Despite moderate awareness of reporting systems, utilization remains poor, reflecting systemic and cultural barriers. These findings highlight the need for targeted educational interventions and system improvements to enhance medication safety culture and error reporting practices across different healthcare specialties and experience levels.

## Introduction

Medication errors (MEs) are common across global health systems and are a major cause of preventable patient harm and healthcare expenditure with an estimated cost of US$ 42 billion annually [1]. In acute care hospitals, the reported incidence of medication errors is approximately 6.5% of patient admissions [2]. Such errors most commonly involve the administration of the wrong drug or dose, the use of an incorrect route, or the provision of medication to the wrong patient [2]. The complexity of modern healthcare systems, combined with increasing medication complexity and workload pressures, creates multiple opportunities for errors throughout the medication use process.

The World Health Organization (WHO) ranks medication-related errors among the top contributors to preventable adverse events, with substantial mortality rates especially in low- and middle-income countries (LMICs) [3]. Ghana faces a heightened medication safety challenge due to inadequate infrastructure, workforce shortages, disrupted workflows, poor training, and a weak culture of reporting, resulting in an undue burden of medication-related harm [4].

Healthcare professionals such as physicians, nurses, and pharmacists, are central to ensuring medication safety, as they are actively engaged at different stages of the medication-use process, namely prescribing, dispensing, and administration [1]. Nurses, particularly, have a unique role and responsibility in medication administration, being frequently the final person to check that medication is correctly prescribed and dispensed before such medications are administered to patients [5].

Understanding healthcare professionals' knowledge of medication errors, their adherence to safety practices, and their engagement with error reporting systems is fundamental to developing effective prevention strategies [6,7]. Medication error reporting systems foster transparency, encourage professionals to report incidents and identify systemic vulnerabilities [8,9].

Previous studies have linked medication errors to both individual factors, including knowledge deficits, and system-level issues, such as poor communication [10,11]. Globally, unsafe medication practices are a leading cause of avoidable patient harm, with most errors occurring during administration [12,13]. Yet, evidence on healthcare professionals' knowledge of error categories and their reporting practices remains limited, particularly in developing health systems. Robust reporting systems are critical to understanding and preventing errors. Addressing gaps between knowledge and practice, and identifying factors that influence reporting awareness and use, are therefore essential to strengthening medication safety culture and reducing error rates.

This multicenter study aims to evaluate healthcare professionals' knowledge of medication error categories and sources, examine their medication administration practices, including adherence to safety protocols, as well as assessing awareness and attitudes toward institutional error reporting systems, and identify gaps between knowledge and practice. By identifying knowledge gaps and practice variations across different professional categories and demographic groups, this research seeks to provide evidence-based recommendations for improving medication safety in healthcare settings in Ghana.

## Methodology

### Study design and setting

A cross-sectional, multicenter design was conducted among healthcare professionals working in tertiary healthcare facilities in Ghana. The various facilities included were the Cape Coast Teaching Hospital (CCTH), Komfo Anokye Teaching Hospital (KATH), Tamale Teaching Hospital (TTH), Sunyani Teaching Hospital (STH), and the Ho Teaching Hospital (HTH). A simple random sampling was used to select five out of the seven teaching hospitals in Ghana to capture diversity in geography and healthcare delivery across Ghana.

### Participants and sampling

The target population included doctors, nurses, and pharmacists directly involved in medication-related tasks in the selected facilities. A total of 2,078 healthcare professionals were recruited. Within each facility, participants were selected using stratified random sampling by profession. Professional categories served as strata to ensure adequate and proportional representation of each group across all sites.

Sample size calculation was based on an estimated population proportion of 50% for awareness of medication error reporting systems, with a 95% confidence level and 3% margin of error. The sample size was calculated using the standard formula for estimating a single proportion in a cross-sectional study as described in other studies [14,15].

$$n = Z^2 p(1-p)/d^2$$

For a confidence level: 95% (Z = 1.96), Estimated proportion: 50% (p = 0.5), and Margin of error: 3% (d = 0.03)

$$n = (1.96)^2 \times 0.5 \times (1-0.5) / (0.03)^2$$

Therefore n= 1,067 participants (minimum), with 15% non-response adjustment, n=1,255.

To ensure equitable representation from all five teaching hospitals, a proportional allocation method was used based on the population size of each facility. A total of 620 respondents were recruited from Komfo Anokye Teaching Hospital, 408 from the Cape Coast Teaching Hospital, 500 from Tamale Teaching Hospital, 250 from Sunyani Teaching hospital, and 300 from Ho Teaching Hospital. A final target of 2078 participants were therefore used to enable subgroup analyses. A response rate between 82–87% was observed across the various study centers.

## Data collection

Data were collected using a structured, pre-tested questionnaire, adapted to the Ghanaian clinical context. The structured questionnaire comprised three sections: (i) knowledge of medication errors (categories, sources, and contributory factors), (ii) attitudes and awareness towards reporting medication errors, and (iii) medication administration practices. Knowledge items were scored 1 for correct and 0 for incorrect or 'not sure' responses. Attitude items were scored on a dichotomous (Yes/No) or Likert-type scale, where higher scores indicated positive attitudes towards reporting. Practice items were rated on a five-point frequency scale (1 = never to 5 = always), with higher scores indicating safer medication practices. The classification of items and scoring criteria are illustrated in Appendix 1 as supporting information (S1 File).

The questionnaires were administered with the help of trained research assistants facilitating data collection across wards and departments. Data was collected from the various healthcare facilities as follows: CCTH from September 1, 2024, to November 30, 2024; TTH from January 1 to March 15, 2025; STH from February 20- April 20, 2025; HTH from March 30- May 30, 2025; and from KATH from May 20 to July 30, 2025. Recruitment for the study was done in a ward setting, where eligible healthcare professionals were approached during their duty hours. Participation in the study happened freely, without any involvement of supervisors in participant selection or data collection. Written informed consent was obtained from all participants, and no incentives or penalties were associated with participation or non-participation.

## Content validity and pilot testing

The questionnaire was developed following a review of the literature on medication errors and patient safety and was informed by established frameworks, including the National Coordinating Council for Medication Error Reporting and Prevention (NCC MERP) taxonomy [16]. Items were adapted to reflect the local clinical context and scope of practice of healthcare professionals in Ghana.

The questionnaire underwent content validity assessment by a panel of five experts in medication safety, including pharmacists, nurses, and physicians with expertise in patient safety research. The content validity index (CVI) exceeded 0.80 for all sections. A pilot study was conducted with 50 healthcare professionals not included in the main study to assess questionnaire clarity, completion time, and preliminary psychometric properties. For knowledge items, responses were coded as "correct" or "incorrect" based on concordance with standard definitions of medication errors as described in international patient safety guidelines. Internal consistency reliability was assessed using Cronbach's alpha. The knowledge, attitude, and practice subscales demonstrated acceptable internal consistency, with Cronbach's alpha values of 0.89, 0.69, and 0.94 respectively.

## Measurements

For descriptive interpretation, knowledge items were categorized based on the proportion of respondents who provided correct responses. Consistent with prior KAP studies, items with ≥75% correct responses were interpreted as reflecting *high knowledge*, those with 50–74% as *moderate knowledge*, and those with <50% as *low knowledge*. These thresholds were used solely for interpretive purposes and not for individual-level scoring.

## Data analysis

Data were entered into a secure database and analyzed using STATA version 14. Descriptive statistics including frequencies, proportions, means, and standard deviation were computed for all variables. Poisson regression models with robust standard errors were used to estimate prevalence ratios (PRs) and 95% confidence intervals for factors associated with awareness. This approach was preferred over logistic regression because the outcome on awareness was common (>10%), and PRs provide a more interpretable measure of association [17]. Subsequently, all exposure variables were included together in a single model, so that the effect of each variable was estimated while accounting for the others. All

exposure variables were included in the multivariable model based on theoretical relevance and prior evidence, irrespective of their statistical significance in bivariate analysis [18]. Multicollinearity among predictors was assessed using variance inflation factors (VIFs). The VIF values ranged from 1.01 to 1.66, with a mean VIF of 1.27, indicating no evidence of problematic multicollinearity.

Socio-demographic characteristics, including age, marital status, education level, professional category, specialty, and years of practice, were included as covariates in the Poisson regression model to assess factors associated with medication error reporting awareness. A p-value < 0.05 was considered statistically significant.

## Ethical considerations

Ethical approval was obtained from the Institutional Review Board of the University of Cape Coast with ID number (UCCIRB/EXT/2024/024), and from all participating hospitals. Participation was voluntary, and informed written consent was obtained from all respondents. Confidentiality of responses was assured, and no identifying information was included in the dataset.

## Results

### Sociodemographic characteristics of respondents

The study comprised 2,078 healthcare professionals (Table 1), with the majority (51.1%) aged between 30–39 years, indicating that the workforce was predominantly young to middle-aged. Females constituted a larger proportion of the sample than males (60.5% vs 39.5%). In terms of educational attainment, diploma (40.9%) and bachelor's degree holders (39.1%) formed the largest groups.

The sample was predominantly Christian (71.4%) compared to Muslims (23.9%). Nurses constituted the largest professional category (67.3%), while pharmacists/pharmacy technicians were the least (11.4%).

### Knowledge on categories and sources of medication errors

Healthcare professionals demonstrated variable knowledge across different medication error categories as shown in supplementary material (Supplementary Table S3) in S2 File. Recognition was highest for fundamental medication errors such as wrong dose (85.5%), wrong drug (82.1%), wrong patient (81.5%), and wrong route of administration (79.7%).

However, knowledge was lower for more complex or system-related error categories. Awareness of drug shortages as a medication error was limited (44%), and recognition of errors related to drug omission, missing dates or signatures, and clinical judgment were moderate. Communication-related sources of medication errors were generally well recognized, including misunderstood verbal orders and unclear prescribing instructions, while system contributors were less recognized.

### Aggregated knowledge levels on medication error categories

Table 2 describes the aggregate knowledge levels of participants on categories of medication errors. Using predefined interpretive thresholds, high knowledge (≥75% correct responses) was observed for commonly recognized medication error categories such as wrong dose and wrong drug, whereas moderate knowledge (50–74%) was observed for documentation-related errors, and low knowledge (<50%) for drug shortages and omission errors. Detailed item-level responses are presented in Supplementary Table S3 in S2 File.

### Individual and systems contributing factors that can contribute to medication errors

Healthcare professionals demonstrated generally high recognition of individual contributory factors to medication errors (Supplementary Table S4) in S2 File. Negligence or inattentiveness (83.9%), failure to follow protocols (81.2%), and

**Table 1. Frequency distribution of sociodemographic factors of participants.**

| Characteristics | Frequency (n) | Percentage (%) |
|---|---|---|
| **AGE (YRS)** | | |
| 20-29 | 684 | 33.5 |
| 30-39 | 1043 | 51.1 |
| 40-49 | 224 | 11 |
| 50-59 | 74 | 3.6 |
| ≥60 | 16 | 0.8 |
| **GENDER** | | |
| Male | 781 | 39.5 |
| Female | 1195 | 60.5 |
| **MARITAL STATUS** | | |
| Married | 838 | 41.3 |
| Single | 1120 | 55.2 |
| Divorced | 50 | 2.47 |
| Widow/widower | 20 | 0.99 |
| **EDUCATIONAL LEVEL** | | |
| Diploma | 830 | 40.9 |
| BSc | 794 | 39.1 |
| MSc/MPhil | 114 | 5.6 |
| General Practitioner | 156 | 7.7 |
| Specialist | 107 | 5.3 |
| Associate Degree | 29 | 1.4 |
| **RELIGION** | | |
| Christian | 1494 | 74.1 |
| Muslim | 482 | 23.9 |
| Traditional | 32 | 1.6 |
| Other | 8 | 0.4 |
| **PROFESSIONAL CATEGORY** | | |
| Doctor | 430 | 21.3 |
| Nurse | 1358 | 67.3 |
| Pharmacist/Pharmacist Technician | 231 | 11.4 |
| **SPECIALTY** | | |
| Medical/general medicine | 599 | 32.5 |
| Surgical | 393 | 21.3 |
| Emergency | 215 | 11.7 |
| OPD | 99 | 5.4 |
| Pediatric | 116 | 6.3 |
| Psychiatry | 109 | 5.9 |
| ICU | 79 | 4.3 |
| Obstetrics and Gynecology (O & G) | 235 | 12.7 |
| **YEARS OF EXPERIENCE** | | |
| 1–3 | 899 | 46.1 |
| 4–6 | 605 | 31.1 |
| 7–9 | 258 | 13.2 |
| >=10 | 186 | 9.5 |

Denominators differ due to item non-response.

**Table 2. Descriptive statistics of aggregated knowledge levels on medication error categories.**

| Knowledge category | Definition | Proportion of respondents (%) |
|---|---|---|
| High knowledge | ≥75% correct identification | 78 |
| Moderate knowledge | 50–74% correct identification | 62 |
| Low knowledge | <50% correct identification | 46 |

inappropriate communication (79.2) were among the most frequently identified individual factors. System-related factors were also widely recognized. Participants commonly identified unclear communication orders (79.1%), inappropriate work locations (73.8%), and lack of access to protocols 74.7%) as contributors to medication errors. Overall, the findings indicate a high level of awareness among healthcare professionals regarding both individual and system-related contributory factors to medication errors.

## Adherence to basic precautions during medication administration processes

Fig 1 illustrates adherence to basic medication administration precautions. Medication administration safety practice adherence showed significant variation across different precautionary measures. Compliance with patient identity verification was generally high, with most respondents (86.6%) indicating that they routinely verified patient identity before administering medications. However, adherence to several other safety precautions was variable. A considerable proportion of participants (37.4%) reported preparing medications for multiple patients simultaneously.

Labeling practices were relatively better, although some participants reported infrequent labeling of syringes or medication preparations. In addition, a notable proportion of respondents indicated that they sometimes administered

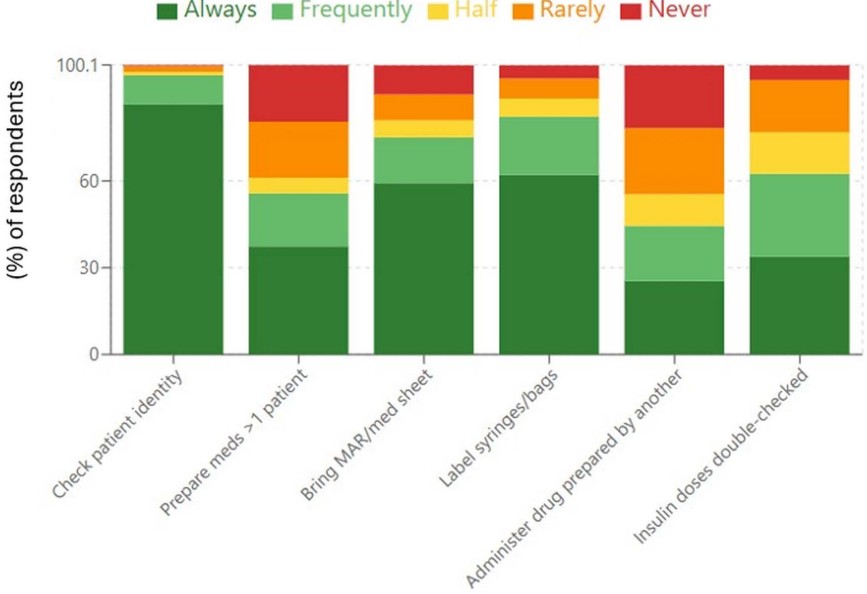

**Fig 1. Frequency distribution of adherence to basic medication administration practices among healthcare professionals (%).** Bars represent the percentage of respondents who reported adherence, indicated as "always, frequently, half of the time, rarely, and never." Percentages are calculated using the total number of respondents for each item (N = 2,015).

medications prepared by another person. Compliance with double-checking procedures for insulin dosing was moderate, with many respondents indicating inconsistent adherence to this precaution.

### Knowledge and practices towards medication errors reporting systems

Awareness of medication error reporting systems was moderate, with most participants (62.6%) knowing about such systems, leaving more than one-third unaware of reporting mechanisms (Fig 2). This awareness gap represents a significant barrier to error reporting and learning from incidents. Actual completion of incident forms was substantially lower, with the minority (29.7%) having completed an incident report, indicating a large gap between awareness and action. Knowledge of how to access reporting forms was also limited, with only about half (46.8%) of participants indicating familiarity with the process.

Formal reporting of medication errors was generally infrequent. Only a minority (23.6%) of respondents reported consistently using incident reporting forms, while a notable proportion indicated that they had never used formal reporting mechanisms.

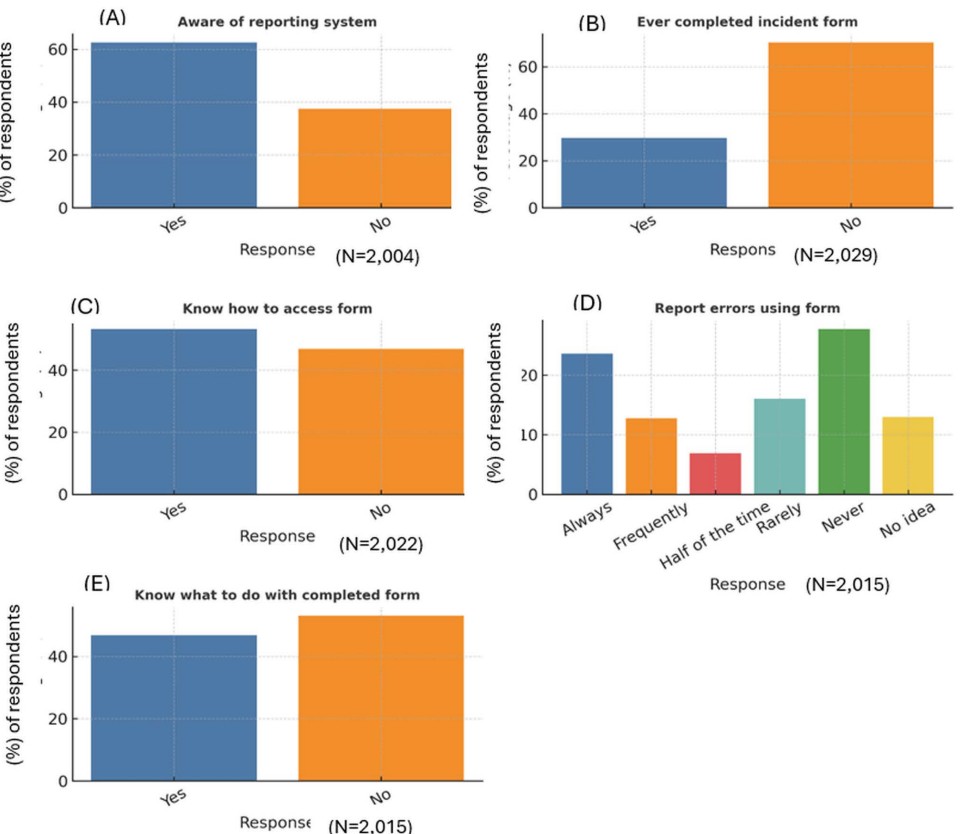

**Fig 2. Distribution of healthcare professionals responding to items on awareness and utilization of the medication incident reporting system.** Bars represent the percentage (%) of respondents selecting each response option for knowledge and utilization items related to the incident reporting system. Graphs A, B, C, D, and E indicate participants' awareness of reporting, ever completed incident form, how to access incident form, report errors using incident forms, and, what to do with completed incident forms respectively.

### Knowledge–attitude–practice (KAP) pattern in medication error reporting

The results in Fig 3 illustrates the Knowledge–Attitude–Practice (KAP) pattern in medication error reporting among health-care professionals. While knowledge of medication error categories was generally high, this was not matched by corresponding attitudes and reporting practices, demonstrating a progressive decline from knowledge to practice.

### Factors associated with awareness of medication error reporting systems

Table 3 presents the results of factors associated with awareness of medication error reporting systems among healthcare professionals. Compared with participants aged 20–29 years, awareness of medication error reporting systems significantly increased with increasing age of respondents: for example, those aged 30–39 years (PR = 1.15; 95% CI 1.03–1.27; p = 0.011), demonstrated significantly higher awareness. Educational attainment was also associated with awareness. Participants with higher degrees such as an MSc/MPhil had higher awareness (PR = 1.23; 95% CI 1.07–1.41; p = 0.003) compared with those holding an associate degree. Professional category and years of experience were significant predictors. Overall, the model suggests that higher educational attainment, moderate work experience, and professional role (particularly pharmacists) were independent predictors of awareness of medication error reporting systems.

## Discussion

This multicenter study offers valuable insights into the knowledge, practices, and attitudes of healthcare professionals toward medication errors (MEs) and reporting systems across Ghana's five major teaching hospitals. The findings reveal both strengths and significant gaps in medication safety awareness and implementation across different professional categories and demographic groups. While awareness of common error types and sources was notably high, and foundational safety practices such as patient identity checks were widely followed, critical gaps remain. Unsafe practices persisted, as only 37.4% always avoided preparing medications for multiple patients at once, while merely 33.9% routinely double-checked insulin doses.

The study demonstrated that healthcare professionals possess strong foundational knowledge of common medication error categories, particularly the traditional "wrong patient, wrong drug, wrong dose, and wrong route" errors that form the core of medication safety education. This aligns with the established emphasis on the 'five rights' of medication administration that has been standard in nursing education and safety training [5]. The high recognition rates for these

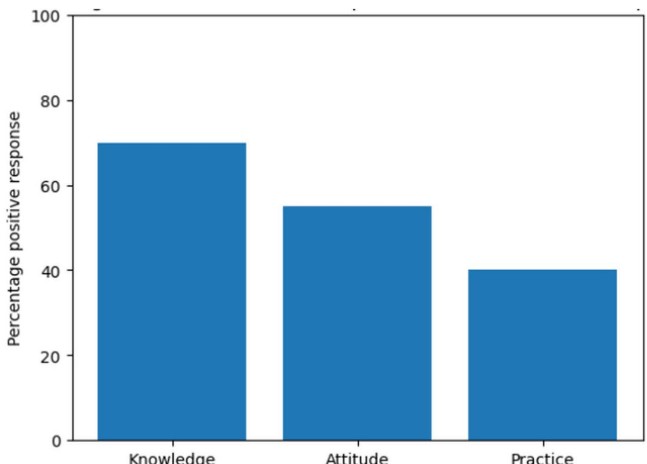

**Fig 3. Distribution of Knowledge–Attitude–Practice (KAP) pattern in medication error reporting among healthcare professionals (%).**

**Table 3. Poisson regression model examining factors associated with awareness of medication error reporting among healthcare professionals. (N = 1,988).**

| Characteristics | cPR (95%CI) | P value | aPR (95% CI) | p value |
|---|---|---|---|---|
| **Age group (yrs)** | | | | |
| 20-29 | 1 | | 1 | |
| 30-39 | 1.01 (0.94-1.10) | 0.750 | 1.15 (1.03-1.27) | 0.011 |
| 40-49 | 1.11 (0.98-1.24) | 0.080 | 1.19(1.02-1.34) | 0.025 |
| 50-59 | 0.91 (0.73-1.14) | 0.410 | 1.32 (1.11-1.57) | 0.001 |
| 60 or more | 1.12 (0.80-1.57) | 0.500 | 1.34 (1.0-1.90) | 0.054 |
| **Marital status** | | | | |
| Divorced | 1 | | 1 | |
| Married | 1.06 (0.84-1.33) | 0.620 | 1.01 (0.93-1.11) | 0.770 |
| Single | 0.98 (0.77-1.22) | 0.830 | 0.88 (0.68-1.15) | 0.350 |
| Widowed | 1.02 (0.68-1.54) | 0.910 | 0.97 (0.61-1.55) | 0.890 |
| **Educational level** | | | | |
| Associate degree | 1 | | 1 | |
| BSc | 0.98 (0.73-1.32) | 0.900 | 1.02 (0.93-1.12) | 0.705 |
| Diploma | 0.99 (0,74-1.34) | 0.970 | 0.98 (0.82-1.17) | 0.822 |
| General Practitioner | 0.86 (0.62-1.20) | 0.380 | 0.83 (0.69-0.99) | 0.044 |
| MSc/MPhil | 0.96 (0.69-1.34) | 0.820 | 1.23 (1.07-1.41) | 0.003 |
| Specialist | 0.96 (0.69-1.34) | 0.820 | 0.61 (0.33-1.11) | 0.110 |
| **Professional category** | | | | |
| Doctor | 1 | | 1 | |
| Nurse | 1.04 (0.35-0.96) | 0.350 | 0.93 (0.83-1.03) | 0.181 |
| Pharmacist | 0.95 (0.47-0.83) | 0.470 | 1.17 (1.03-1.32) | 0.012 |
| **Specialty** | | | | |
| Emergency | 1 | | 1 | |
| ICU | 0.79 (0.62-1.0) | 0.050 | 0.97 (0.87-1.07) | 0.530 |
| Medical/general medicine | 0.93 (0.83-1.04) | 0.200 | 1.03 (0.91-1.16) | 0.690 |
| O&G/ delivery suite | 0.84 (0.72-0.97) | 0.020 | 1.01 (0.85-1.19) | 0.930 |
| OPD | 0.89 (0.75-1.07) | 0.250 | 1.01 (0.88-1.16) | 0.900 |
| Pediatric | 0.89 (0.75-1.06) | 0.210 | 0.97 (0.82-1.15) | 0.760 |
| Psychiatry | 0.96 (0.81-1.14) | 0.650 | 0.90 (0.71-1.12) | 0.350 |
| Surgical | 0.97 (0.86-1.10) | 0.580 | 0.87 (0.75-1.01) | 0.060 |
| **Years of experience** | | | | |
| 1–3 | 1 | | 1 | |
| 4–6 | 1.03 (0.95-1.11) | 0.540 | 1.12 (1.11-1.34) | <0.001 |
| 7–9 | 0.98 (0.87-1.10) | 0.700 | 1.10 (0.97-1.26) | 0.132 |
| 10 or more | 0.94 (0.82-1.08) | 0.390 | 1.14 (0.98-1.34) | 0.096 |

Estimates are presented as crude prevalence ratios (cPRs) and adjusted prevalence ratios (aPRs) with 95% confidence intervals (CIs) derived from a Poisson regression model with robust standard errors. The multivariable Poisson regression model included age, marital status, education level, professional category, specialty, and years of practice as covariates.

fundamental error types (>80%) suggest that basic medication safety concepts are well-integrated into professional training programs.

However, concerning knowledge gaps emerged for more complex or system-related error categories. The particularly low recognition of drug shortage as a medication error (44.0%) indicates limited understanding of supply chain issues as safety concerns. This finding is significant given increasing global medication supply challenges and their impact on patient safety [19]. Healthcare professionals may not recognize that unavailability of prescribed medications can lead to treatment delays, inappropriate substitutions, or therapeutic failures that constitute medication errors.

The moderate recognition of documentation-related errors and omission errors suggests areas where enhanced training could improve safety awareness. Given that nurses provide a vital function in detecting and preventing errors that occurred in prescribing and transcription phase [19], improved recognition of this error types could strengthen the healthcare system's overall error prevention capability.

The practice patterns revealed significant inconsistencies in safety behavior adherence, with patient identification showing excellent compliance (86.6% always performed) while other critical safety practices showed substantial gaps. The concerning finding that only 25.4% of professionals always avoided administering medications prepared by others represents a fundamental breach of medication safety principles and individual accountability mechanisms. Previous research has identified that fewer than 50% of healthcare professionals consistently implement certain safety practices [14], and our findings align with these concerning patterns. The practice of preparing medications for multiple patients simultaneously (with only 37.4% consistently preparing for single patients) creates substantial error risks through increased complexity and reduced focus.

These practice gaps may be associated with workplace pressures, inadequate staffing, or normalization of deviance where unsafe practices become accepted over time. The largest proportion of medication errors occur during medication administration, with nurses playing a significant role in both occurrence and prevention of medication administration errors [6].

The substantial gap between reporting system awareness (62.6%) and actual utilization (29.7% ever completed a form) represent one of the most critical findings of this study. While error reporting systems foster transparency and facilitate identification of trends and root causes of errors, their effectiveness depends on active professional engagement [20]. This utilization gap suggests significant barriers to reporting that extends beyond simple awareness issues.

The finding that only 23.6% of professionals always use formal reporting systems for errors indicates that most medication errors are likely to remain unreported through official channels. This underreporting may constrain healthcare systems' capacity to identify systematic problems, track error trends, and implement preventive interventions.

The analysis revealed that awareness of medication error reporting systems increased with age, professional experience, and level of education. This pattern suggests that prolonged exposure to clinical practice and patient safety programs may improve understanding of institutional reporting procedures. The higher awareness among pharmacists compared to doctors and nurses may reflect their central role in medication management and familiarity with pharmacovigilance protocols.

Conversely, lower awareness among general practitioners could be attributed to limited engagement in structured error reporting systems or competing clinical workloads that divert attention from administrative safety processes. The lack of significant variation across specialties indicates that awareness barriers are likely systemic rather than department-specific, consistent with Reason's *Swiss Cheese Model* and Vincent's *London Protocol*, which emphasize organizational and cultural influences on error reporting [18,21].

Through the lens of Vincent's London Protocol, the current findings reflect deficiencies across key domains influencing clinical incidents, including task factors (unclear procedures), individual factors (training and competence), team factors (hierarchical communication), and organizational factors (resource constraints and culture of blame) [18]. Addressing these interconnected layers will require a system-wide safety approach that promotes open communication, non-punitive

reporting, and regular training to strengthen the 'defensive barriers' that prevent harm. Applying these frameworks implies that improving medication error reporting cannot rely solely on individual awareness or willingness to report. Rather, it requires systemic redesign, embedding robust reporting channels, routine feedback, and leadership commitment to safety culture. Integrating Vincent's Protocol can therefore foster an environment where healthcare workers feel supported to report errors as learning opportunities rather than as grounds for punishment.

Taken together, these findings indicate that medication error reporting behavior among healthcare professionals is shaped by the interaction between knowledge, clinical practice, and systemic conditions rather than by awareness alone. While knowledge of traditional medication error categories was high, unsafe practices persisted and formal reporting remained limited, suggesting a disconnect between what healthcare professionals know and what they are able or willing to do in routine practice. Underreporting of medication errors has been attributed to organizational issues such as fear of blame, unstructured systems for reporting medication errors, workload, accountability, repercussions, as well as individual and professional concerns [22]. Viewed through Vincent's London Protocol and Reason's Swiss Cheese Model, this disconnect reflects weaknesses across multiple defensive layers, including individual factors (knowledge gaps in system-level errors), task and team factors (workload pressures and informal workflows), and organizational factors (resource constraints and punitive cultures) that allow multiple latent and active failures to align and impede safe practice and reporting [23]. Within the Ghanaian context, the combined vulnerabilities facilitate the normalization of unsafe practices and discourage formal reporting, allowing errors to pass through weakened system barriers. Consequently, improving medication error reporting requires system-level interventions that strengthen organizational support, promote non-punitive safety cultures, and integrate feedback mechanisms, rather than relying solely on individual knowledge or awareness.

Given Ghana's resource constraints, including workforce shortages, training gaps, and weak health system infrastructure, strengthening the capacity for medication error reporting is both imperative and challenging [4,24]. High patient-to-provider ratios and limited continuous professional development exacerbate error risk and undermine safety systems. To respond effectively, targeted training programs must emphasize not just knowledge but also operational aspects, clarifying what to report, how, and why the need for reporting. Moreover, critical feedback mechanisms must be in place such that respondents will feel their reporting will lead to learning and system improvement but not just blaming them [25,26].

The findings have significant implications for national policy and practice. Within Ghana's health governance framework, both the Ghana Health Service and the Ministry of Health play critical roles in ensuring patient safety through institutional guidelines and workforce training [27,28]. The observed underreporting of medication errors, coupled with inadequate feedback mechanisms, suggests the need to reinforce a non-punitive reporting culture within healthcare facilities. The Ghana Health Service could integrate structured medication error reporting indicators into its Patient Safety and Quality Improvement Strategic Plan, ensuring that incident reporting becomes a learning tool rather than a disciplinary measure. Additionally, the Ministry of Health, through its Continuous Professional Development (CPD) and Health Workforce Policy, can strengthen periodic in-service training on medication safety, error prevention, and reporting protocols across all professional cadres. Embedding such modules in mandatory CPD programs would help harmonize knowledge and attitudes across different staff categories and facilities.

## Strengths and limitations

Our study's strength lies in its multicenter design across diverse regional teaching hospitals, enabling a robust understanding of practices and perceptions in varied clinical environments. Also, another major strength of this study is the high participation rate (82–87%), which enhances the representativeness and internal validity of the findings. Recruitment was conducted using a ward-based approach with voluntary participation and no supervisory involvement, thereby minimizing the risk of coercion. Nevertheless, as with all voluntary surveys, the possibility of non-response bias cannot be completely excluded, particularly if healthcare professionals with greater interest in medication safety were more likely to participate. Additionally, reliance on self-reported data could introduce social desirability and recall biases, potentially underestimating

the true frequency of errors. Future research should consider observational or longitudinal approaches to validate self-reported behaviors and monitor intervention effectiveness. Moreover, this study did not examine interpersonal or cultural dynamics such as hierarchical relationships, intimidation, or teamwork climate among healthcare professionals, nor did it assess institutional training practices. These system-level and behavioral factors are known to influence patient safety culture and medication error reporting but were outside the scope of the quantitative design and dataset used. Future mixed-methods studies are warranted to explore these dimensions in greater depth, including how workplace culture, professional interactions, and opportunities for continuous training, shape reporting behaviors.

## Conclusion

Overall, this study highlights a compelling scenario, as healthcare professionals in Ghana possess sound knowledge about medication errors and reporting systems, but actual reporting behaviors remain deficient, hampered by systemic and cultural roadblocks. Effective progress will require a multi-pronged strategy, encompassing system redesign, cultural change, education, and leadership support, to realize a safer, more transparent medication use environment in Ghana's healthcare system.

The findings highlight critical needs for comprehensive medication safety interventions addressing knowledge gaps, practice standardization, and reporting system enhancement. The study underscores the need for national policy reforms that promote non-punitive error reporting and integrates medication safety into continuing professional development programs.

## Supporting information

**S1 File. Classification of items and scoring criteria.**
(DOCX)

**S2 File. Supplementary Table S3 and Table S4 on medication error knowledge and contributing factors.**
(DOCX)

## Acknowledgments

The authors wish to express their sincere gratitude to all the participants who were available for this study. We also wish to thank the management of the various healthcare facilities for making their hospital accessible for this research.

## Author contributions

**Conceptualization:** Stephen Mensah Arhin.

**Data curation:** Stephen Mensah Arhin.

**Formal analysis:** Stephen Mensah Arhin, Isaac Tabiri Henneh.

**Funding acquisition:** Stephen Mensah Arhin, Kwasi Sobre Nkrumah.

**Investigation:** Stephen Mensah Arhin, Isaac Tabiri Henneh, Ewura Seidu Yahaya.

**Methodology:** Stephen Mensah Arhin, Isaac Tabiri Henneh, Kwasi Sobre Nkrumah, Ewura Seidu Yahaya, Phyllis Elsie Owusu Agyei, Meshack Antwi-Adjei, Martins Ekor.

**Project administration:** Stephen Mensah Arhin, Kwasi Sobre Nkrumah, Martins Ekor.

**Resources:** Stephen Mensah Arhin, Isaac Tabiri Henneh.

**Software:** Isaac Tabiri Henneh, Ewura Seidu Yahaya.

**Supervision:** Stephen Mensah Arhin, Martins Ekor.

**Validation:** Phyllis Elsie Owusu Agyei, Meshack Antwi-Adjei.

**Visualization:** Stephen Mensah Arhin, Isaac Tabiri Henneh, Kwasi Sobre Nkrumah.

**Writing – original draft:** Stephen Mensah Arhin, Isaac Tabiri Henneh, Ewura Seidu Yahaya.

**Writing – review & editing:** Stephen Mensah Arhin, Isaac Tabiri Henneh, Kwasi Sobre Nkrumah, Ewura Seidu Yahaya, Phyllis Elsie Owusu Agyei, Meshack Antwi-Adjei, Martins Ekor.

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
