## [Decision Letter · Decision Letter 0]

6 Nov 2025

PONE-D-25-54066Medication Error Reporting in Ghana: A Multicenter Assessment of Healthcare Professionals’ Knowledge, Attitudes and Practices.PLOS ONE

Dear Dr. Arhin,

Thank you for submitting your manuscript to PLOS ONE. After careful consideration, we feel that it has merit but does not fully meet PLOS ONE’s publication criteria as it currently stands. Therefore, we invite you to submit a revised version of the manuscript that addresses the points raised during the review process.

We look forward to receiving your revised manuscript.

Kind regards,

Douglas Aninng Opoku, MPH

Academic Editor

PLOS ONE

Journal Requirements:

Reviewers' comments:

Reviewer's Responses to Questions

**Comments to the Author**

1. Is the manuscript technically sound, and do the data support the conclusions?

Reviewer #1: Yes

Reviewer #2: Yes

2. Has the statistical analysis been performed appropriately and rigorously? 

Reviewer #1: Yes

Reviewer #2: No

3. Have the authors made all data underlying the findings in their manuscript fully available?

Reviewer #1: Yes

Reviewer #2: No

4. Is the manuscript presented in an intelligible fashion and written in standard English?

Reviewer #1: Yes

Reviewer #2: Yes

5. Review Comments to the Author

Reviewer #1: Results

It is observed that there is no mention of the tabulated results in the result report. Authors are advised to address this so that readers can follow what tabulated results is being reported.

Discussion

Line 327: Authors have to look at that sentence and put it in a way that conveys what they intend to put out. The sentence does not make sense.

Overall Comment

The study is very relevant, and it present a critical health practise concern. However, it would be essential to show or demonstrate the work relationship across the health professionals of these facilities. Was there a case of inferiority complex setting in, or intimidation or outright negligence and lack of facility members not getting upgraded with best drug reporting and administration practices? It would be also essential to probe into if the facilities carry out routine trainings to update the staff on recent practices. The drawback is observed with having the newer staff having the knowledge and skills to report while the older or more experienced members are lagging behind. Such submission is a possible indictment on the facility.

Reviewer #2: Reviewer Comments to Authors

1. Summary

This study examines healthcare professionals’ knowledge, attitudes, and practices regarding medication-error reporting in five Ghanaian teaching hospitals (n = 2,078). Using a validated questionnaire and descriptive plus logistic analyses, the authors identify high theoretical awareness but low utilization of reporting systems. The work addresses a crucial yet under-researched patient-safety issue in low- and middle-income countries and aligns well with WHO’s Medication Without Harm initiative.

2. General Assessment

The manuscript is well organized and clearly written. It contributes valuable multicenter evidence from Ghana, but it requires methodological clarification, analytic transparency, and deeper theoretical contextualization before it can meet PLOS ONE standards of reproducibility and interpretive rigor.

Overall recommendation: Major revision

3. Major Comments

Sampling and representativeness: Clarify how sampling ensured representativeness within each hospital and report participation/response rates and handling of missing data.

Questionnaire details: Provide an appendix or supplementary table showing key items, response options, and scoring for the Knowledge–Attitude–Practice domains. It is not explicitly clear, for example, which responses constitute attitudes and practices.

Statistical modeling: Indicate dependent variable coding, independent variables entered, confounders adjusted for, and model-fit indices such as Hosmer–Lemeshow and pseudo-R².

Interpretation of associations: Acknowledge the cross-sectional design limitation and avoid causal phrasing such as 'led to' or 'resulted in'.

Discussion depth: Integrate Reason’s Swiss Cheese model and/or Vincent’s London Protocol to interpret system and cultural barriers to reporting.

Policy linkage: Discuss how findings could inform Ghana Health Service or Ministry of Health policies on non-punitive reporting and continuing professional development.

Data sharing compliance: Provide an anonymized dataset or repository link to satisfy PLOS ONE data-availability policy.

4. Minor Comments

Replace 'Data was analyzed' with 'Data were analyzed'.

Correct 'responders will t feel' to 'respondents will feel'.

Use 'Obstetrics and Gynecology (O&G)' instead of 'O & G/delivery suits'.

Maintain consistent spelling of behavior.

Ensure uniform capitalization of 'Knowledge, Attitudes, and Practices'.

Verify numerical consistency in tables, as some denominators differ from 2,078.

Simplify dense tables and consider adding a visual summary such as a bar chart of adherence or awareness rates.

5. Strengths to Retain

Large and diverse multicenter dataset.

Robust ethical governance and validation process.

Timely relevance to LMIC patient-safety agendas.

Clear and well-written results section with helpful tables.

6. Summary of Required Revisions

1. Expand methodological transparency regarding sampling, instrument, and regression.

2. Deepen discussion with systems-safety frameworks.

3. Improve clarity, grammar, and table consistency.

4. Upload dataset or repository link.

5. Check reference and format compliance with PLOS ONE style.

When these revisions are addressed, the manuscript will make a significant contribution to global medication-safety literature.

6. PLOS authors have the option to publish the peer review history of their article (what does this mean?). If published, this will include your full peer review and any attached files.

Reviewer #1: **Yes:** Inemesit Okon Ben

Reviewer #2: **Yes:** Aaron Asibi Abuosi

---

## [Author Response · Author response to Decision Letter 1]

14 Nov 2025

The concerns raised by the reviewers and the editor have been addressed and attached in the submission system

---

## [Decision Letter · Decision Letter 1]

26 Dec 2025

PONE-D-25-54066R1Medication Error Reporting in Ghana: A Multicenter Assessment of Healthcare Professionals’ Knowledge, Attitudes and Practices.PLOS One

Dear Dr. Arhin,

Thank you for submitting your manuscript to PLOS ONE. After careful consideration, we feel that it has merit but does not fully meet PLOS ONE’s publication criteria as it currently stands. Therefore, we invite you to submit a revised version of the manuscript that addresses the points raised during the review process.

We look forward to receiving your revised manuscript.

Kind regards,

Douglas Aninng Opoku, MPH

Academic Editor

PLOS One

Journal Requirements:

Reviewers' comments:

Reviewer's Responses to Questions

**Comments to the Author**

1. If the authors have adequately addressed your comments raised in a previous round of review and you feel that this manuscript is now acceptable for publication, you may indicate that here to bypass the “Comments to the Author” section, enter your conflict of interest statement in the “Confidential to Editor” section, and submit your "Accept" recommendation.

Reviewer #1: All comments have been addressed

Reviewer #2: (No Response)

2. Is the manuscript technically sound, and do the data support the conclusions?

Reviewer #1: Yes

Reviewer #2: Yes

3. Has the statistical analysis been performed appropriately and rigorously? 

Reviewer #1: Yes

Reviewer #2: Yes

4. Have the authors made all data underlying the findings in their manuscript fully available?

Reviewer #1: Yes

Reviewer #2: (No Response)

5. Is the manuscript presented in an intelligible fashion and written in standard English?

Reviewer #1: Yes

Reviewer #2: Yes

6. Review Comments to the Author

Reviewer #1: (No Response)

Reviewer #2: (No Response)

7. PLOS authors have the option to publish the peer review history of their article (what does this mean?). If published, this will include your full peer review and any attached files.

Reviewer #1: **Yes:** Inemesit Okon Ben

Reviewer #2: No

---

## [Author Response · Author response to Decision Letter 2]

15 Jan 2026

A point to point response to the reviewer comments has been uploaded as an attached document

---

## [Decision Letter · Decision Letter 2]

12 Feb 2026

PONE-D-25-54066R2Medication Error Reporting in Ghana: A Multicenter Assessment of Healthcare Professionals’ Knowledge, Attitudes and Practices.PLOS One

Dear Dr. Arhin,

Thank you for submitting your manuscript to PLOS ONE. After careful consideration, we feel that it has merit but does not fully meet PLOS ONE’s publication criteria as it currently stands. Therefore, we invite you to submit a revised version of the manuscript that addresses the points raised during the review process.

We look forward to receiving your revised manuscript.

Kind regards,

Douglas Aninng Opoku, MPH

Academic Editor

PLOS One

Journal Requirements:

Additional Editor Comments:

In Table 4, the authors presented only adjusted prevalence ratio without presenting the crude. It is recommended that the authors present both the crude and adjusted and also explain what informed the variable selection for the multivariable model.

Reviewers' comments:

Reviewer's Responses to Questions

**Comments to the Author**

1. If the authors have adequately addressed your comments raised in a previous round of review and you feel that this manuscript is now acceptable for publication, you may indicate that here to bypass the “Comments to the Author” section, enter your conflict of interest statement in the “Confidential to Editor” section, and submit your "Accept" recommendation.

Reviewer #2: All comments have been addressed

2. Is the manuscript technically sound, and do the data support the conclusions?

Reviewer #2: Yes

3. Has the statistical analysis been performed appropriately and rigorously? 

Reviewer #2: Yes

4. Have the authors made all data underlying the findings in their manuscript fully available?

Reviewer #2: Yes

5. Is the manuscript presented in an intelligible fashion and written in standard English?

Reviewer #2: Yes

6. Review Comments to the Author

Reviewer #2: (No Response)

7. PLOS authors have the option to publish the peer review history of their article (what does this mean?). If published, this will include your full peer review and any attached files.

Reviewer #2: **Yes:** Aaron Asibi Abuosi

---

## [Author Response · Author response to Decision Letter 3]

15 Feb 2026

Response to editor comments has been included

---

## [Editor Report · Decision Letter 3]

20 Feb 2026

PONE-D-25-54066R3Medication Error Reporting in Ghana: A Multicenter Assessment of Healthcare Professionals’ Knowledge, Attitudes and Practices.PLOS One

Dear Dr.  Arhin,

Thank you for submitting your manuscript to PLOS ONE. After careful consideration, we feel that it has merit but does not fully meet PLOS ONE’s publication criteria as it currently stands. Therefore, we invite you to submit a revised version of the manuscript that addresses the points raised during the review process.

We look forward to receiving your revised manuscript.

Kind regards,

Douglas Aninng Opoku, MPH

Academic Editor

PLOS One

Journal Requirements:

Additional Editor Comments:

1. There should be consistency in the reporting of p-values. Eg. in Table 4, some of the p-values are in 1, 2 and 3 decimal places. All should be consistent throughout the manuscript

2. Subsequently, all exposure variables were included together in a single multivariate model, so that the effect of each variable was estimated while accounting for the others. Delete the word 'multivariate' from this sentence as authors are not assessing more than one outcome

3. Provide a reference for this statement. This approach was preferred over logistic regression because the outcome on awareness was common (>10) and PRs provide a more interpretable measure of association. All exposure variables were included in the multivariable model based on theoretical relevance and prior evidence

4. Multicollinearity among predictors was assessed using variance inflation factors (VIFs); the mean VIF was 1.27, indicating no evidence of problematic multicollinearity. Report the minimum and maximum values for the variables in the model for the VIF in addition to the mean VIF

---

## [Editor Report · Decision Letter 4]

13 Mar 2026

PONE-D-25-54066R4Medication Error Reporting in Ghana: A Multicenter Assessment of Healthcare Professionals’ Knowledge, Attitudes and Practices.PLOS One

Dear Dr. Arhin,

Thank you for submitting your manuscript to PLOS ONE. After careful consideration, we feel that it has merit but does not fully meet PLOS ONE’s publication criteria as it currently stands. Therefore, we invite you to submit a revised version of the manuscript that addresses the points raised during the review process.

We look forward to receiving your revised manuscript.

Kind regards,

Douglas Aninng Opoku, MPH

Academic Editor

PLOS One

Journal Requirements:

Additional Editor Comments :

I appreciate the authors for responding to the comments that were raised in my last review. However, not all of them have been addressed. Especially, Table 4 still has some of the p-values presented in 3 decimals while some also are presented in 2. Variable "Age (9yrs)" is an error and should be corrected by the authors. Again, 'Age' in all the tables should be changed to "Age group (years)". The results write-up also need to be re-written. The current format report almost everything in the Tables and should not be so. The Table is there as a reference and authors should not report everything in it in the write up.

---

## [Author Response · Author response to Decision Letter 5]

16 Mar 2026

The manuscript has been revised to include the editor's suggestions

---

## [Editor Report · Decision Letter 5]

6 Apr 2026

PONE-D-25-54066R5Medication Error Reporting in Ghana: A Multicenter Assessment of Healthcare Professionals’ Knowledge, Attitudes and Practices.PLOS One

Dear Dr. Arhin,

Thank you for submitting your manuscript to PLOS ONE. After careful consideration, we feel that it has merit but does not fully meet PLOS ONE’s publication criteria as it currently stands. Therefore, we invite you to submit a revised version of the manuscript that addresses the points raised during the review process.

We look forward to receiving your revised manuscript.

Kind regards,

Douglas Aninng Opoku, MPH

Academic Editor

PLOS One

Journal Requirements:

Additional Editor Comments:

1. The results write up still needs to be improved. Add the estimates in bracket. Eg majority (23.3%) of the participants were males etc..

2. Remove all the totals in Table 1.

---

## [Author Response · Author response to Decision Letter 6]

7 Apr 2026

All comments have been responded to

---

## [Editor Report · Decision Letter 6]

12 Apr 2026

PONE-D-25-54066R6Medication Error Reporting in Ghana: A Multicenter Assessment of Healthcare Professionals’ Knowledge, Attitudes and Practices.PLOS One

Dear Dr. Arhin,

Thank you for submitting your manuscript to PLOS ONE. After careful consideration, we feel that it has merit but does not fully meet PLOS ONE’s publication criteria as it currently stands. Therefore, we invite you to submit a revised version of the manuscript that addresses the points raised during the review process.

We look forward to receiving your revised manuscript.

Kind regards,

Douglas Aninng Opoku, MPH

Academic Editor

PLOS One

Journal Requirements:

Additional Editor Comments :

Please take your time and address the concerns I raised in my last review. You only addressed the concerns I raised partially. Apart from removing the total from Table 1, the other comment was not addressed. Re-write your results section and do not state everything in the table. Also just mentioning majority without stating the percentage is not enough.

---

## [Editor Report · Decision Letter 7]

27 Apr 2026

Medication Error Reporting in Ghana: A Multicenter Assessment of Healthcare Professionals’ Knowledge, Attitudes and Practices.

PONE-D-25-54066R7

Dear Dr. Arhin,

We’re pleased to inform you that your manuscript has been judged scientifically suitable for publication and will be formally accepted for publication once it meets all outstanding technical requirements.

Kind regards,

Douglas Aninng Opoku, MPH

Academic Editor

PLOS One
---

## [Editor Report · Acceptance letter]

PONE-D-25-54066R7

PLOS One

Dear Dr. Arhin,

I'm pleased to inform you that your manuscript has been deemed suitable for publication in PLOS One. Congratulations! Your manuscript is now being handed over to our production team.

Kind regards,

on behalf of

Dr. Douglas Aninng Opoku

Academic Editor

PLOS One